# Learning Pyramid Representations from Gigapixel Histopathological Images

## Abstract

Whole slide images (WSIs) pose fundamental computational challenges due to their gigapixel resolution and the sparse distribution of informative regions. Existing approaches often treat image patches independently—discarding spatial structure—or reshape them in ways that distort spatial context, thereby obscuring the hierarchical pyramid representations intrinsic to WSIs. We introduce Sparse Pyramid Attention Networks (SPAN), a hierarchical framework that preserves spatial relationships while efficiently allocating computation to informative regions. SPAN constructs multi-scale representations directly from single-scale inputs, enabling precise WSI modeling without sacrificing efficiency. We demonstrate SPAN's versatility through two variants: SPAN-MIL for slide classification and SPAN-UNet for segmentation. Comprehensive evaluations across multiple public datasets show that SPAN captures the hierarchical structure and contextual relationships that existing methods fail to model. Our results provide clear evidence that architectural inductive biases and hierarchical representations enhance both slide-level and patch-level performance. By overcoming long-standing computational barriers, SPAN establishes a new paradigm for computational pathology and reveals foundational design principles for large-scale medical image analysis.

## 1 Introduction

Whole Slide Images (WSIs) have become indispensable in modern digital pathology. These high-resolution scans, typically derived from Hematoxylin and Eosin (H&E)-stained tissue samples, allow precise identification of cellular structures and abnormalities. By digitizing histopathological slides, WSIs enable pathologists to analyze tissue samples across multiple scales, ranging from high-level tissue architecture to fine-grained cellular morphology, thereby supporting more accurate and efficient diagnoses. Beyond manual examination, WSIs facilitate computer-aided diagnosis (Campanella et al., 2019; Abels et al., 2019) and serve as the foundation for a variety of computational pathology tasks. At the *patch level*, localized problems such as nuclei segmentation (Lou et al., 2024; Lin et al., 2024) and tissue classification (Veeling et al., 2018) can be effectively addressed using standard computer vision methods, since the scale is manageable and the regions of interest are well defined.

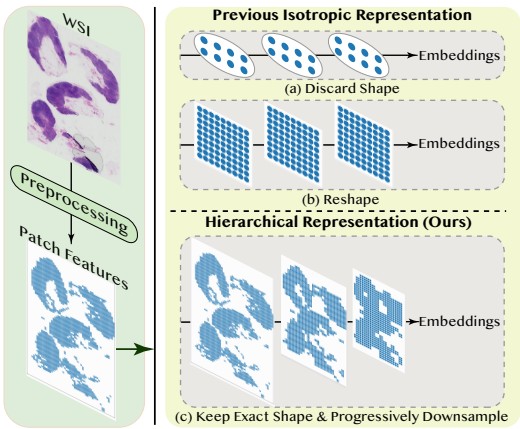

Figure 1: Left: A WSI is preprocessed by patch tiling and feature extraction. Right: (a) Patches treated as i.i.d. samples. (b) Patches reshaped into squares or flattened. (c) Patches preserved in their original shapes and progressively merged.

In contrast, *slide-level* analysis presents fundamentally different computational challenges due to the gigapixel scale of WSIs and the sparse and irregular distribution of informative regions (Lu et al., 2021). Key slide-level tasks include tumor detection, subtyping, and grading (Brancati et al., 2022; Bejnordi et al., 2017; Network et al., 2012; 2014), which rely on histologically grounded labels

with relatively low noise. More recently, tasks such as biomarker prediction (Coudray et al., 2018; Jin et al., 2024; El Nahhas et al., 2024) and survival prediction (Chen et al., 2021; Li et al., 2023) have drawn increasing interest. Biomarker prediction requires linking visual features to genetic alterations, while survival prediction—although inherently a regression problem—is often framed as classification via discretized survival times. In these settings, labels are derived from clinical or genomic data and may not correspond directly to visual cues, making the discovery of non-obvious histopathological patterns especially challenging.

Because WSIs often exceed billions of pixels, direct end-to-end analysis is computationally infeasible with conventional vision models. Moreover, large regions of background or non-diagnostic content necessitate approaches that can efficiently focus on informative tissue. A widely adopted strategy divides WSIs into smaller patches for independent analysis (Bejnordi et al., 2017; Campanella et al., 2019), treating them as i.i.d. samples (Campanella et al., 2019; Lu et al., 2021) (Fig.1, Top). Alternatively, some methods reshape sparse patches into dense square grids to enable convolutional processing(Shao et al., 2021; Tang et al., 2024) (Fig.1, Middle). However, this reshaping disrupts true spatial relationships, since WSI regions are inherently irregularly distributed. Both strategies either ignore or distort the hierarchical spatial organization of WSIs, which risks discarding critical diagnostic information. Our approach instead constructs hierarchical representations that preserve exact spatial relationships and capture multi-scale context (Fig.1, Bottom), directly addressing these limitations.

Recent advances in deep learning, particularly Transformer-based models, demonstrate remarkable success in modeling long-range dependencies in both language (Devlin et al., 2018; Liu, 2019) and vision (Dosovitskiy et al., 2021; Hatamizadeh et al., 2024; Darcet et al., 2024). However, applying them directly to WSIs remains infeasible: The quadratic complexity of vanilla attention is prohibitive at the gigapixel scale (Vaswani et al., 2017). Although sparse and hierarchical attention variants (Beltagy et al., 2020; Zaheer et al., 2020; Wang et al., 2021; Liu et al., 2021) mitigate this in dense, regularly shaped data, they are poorly suited for WSIs, where informative content is both sparse and irregular. Consequently, WSI-specific Transformer models attempt to circumvent this mismatch by reshaping sparse regions into dense grids. For example, TransMIL (Shao et al., 2021) relies on re-squaring with Nyström attention and [CLS] tokens, while others introduce region attention after dense reshaping (Tang et al., 2024). These approaches inevitably distort positional information and restrict modeling to isotropic representations, failing to exploit the hierarchical structures that have proven vital in general computer vision.

To address these challenges, we propose the Sparse Pyramid Attention Network (SPAN), a sparse-native framework for WSI analysis. SPAN preserves exact spatial information while enabling hierarchical operations such as shifted-window attention and multi-scale feature downsampling, bridging the gap between general computer vision architectures and WSI-specific needs. Its design integrates two complementary modules: the Spatial-Adaptive Feature Condensation (SAC) module, which progressively builds hierarchical representations by condensing informative regions, and the Context-Aware Feature Refinement (CAR) module, which captures complex local and global dependencies at each scale. Together, they direct computation toward diagnostically relevant areas and, for the first time, make pyramid-style architectures from general vision effective for WSI analysis.

We validate SPAN across multiple public datasets (Network et al., 2012; 2014; Aresta et al., 2019; Brancati et al., 2022; Bejnordi et al., 2017; Bandi et al., 2018) on classification and segmentation tasks. Experiments demonstrate that SPAN consistently outperforms state-of-the-art methods by capturing spatial and contextual information more effectively. Our main contributions are:

- A sparse computational framework that preserves spatial relationships in WSIs, enabling the direct use of hierarchical vision techniques.

- The SPAN architecture with SAC and CAR modules, which jointly build multi-scale representations through spatial-adaptive condensation and contextual refinement, supporting flexible task-specific variants.

- Comprehensive evaluations demonstrate that embedding *hierarchical and sparsity-aware inductive biases* into the architecture substantially enhances the representation learning on gigapixel histopathological images.

## 2 Preliminary: Whole Slide Image Analysis

### 2.1 Isotropic Paradigms

WSIs inherently possess a hierarchical structure, enabling pathologists to examine tissue samples across multiple magnification levels. This multi-scale nature of WSIs underscores the importance of capturing and integrating information from different scales for accurate analysis. However, most existing computational methods fail to fully exploit this characteristic, operating in an isotropic manner—maintaining constant spatial resolution and feature dimensions throughout processing, without the hierarchical downsampling that enables efficient multi-scale reasoning. Mainstream WSI analysis techniques treat patches as independent and identically distributed (i.i.d.) samples, completely disregarding spatial relationships (Ilse et al., 2018; Lu et al., 2021; Li et al., 2021; Zhang et al., 2022; Tang et al., 2023). Attention-based Multiple Instance Learning (ABMIL) (Ilse et al., 2018) serves as a foundational approach, aggregating patch-level features for slide-level prediction. Extensions like CLAM (Lu et al., 2021) and DTFD-MIL (Zhang et al., 2022) introduce additional losses or training strategies but still neglect spatial context.

Even methods that attempt to incorporate spatial information remain fundamentally isotropic while introducing additional distortions. TransMIL and its variants (Shao et al., 2021; Tang et al., 2024) reshape sparse patches into dense 2D grids, while other approaches (Yang et al., 2024; Zheng et al., 2025; Fillioux et al., 2023) flatten patches into sequences. Both strategies forcibly convert sparse inputs into dense representations, also distorting real positional relationships by artificially connecting non-adjacent patches. Crucially, all these approaches process patches at uniform resolution with fixed feature dimensions throughout the network, failing to leverage hierarchical modeling capabilities that have proven crucial in general computer vision tasks. Consequently, WSI analysis has been unable to benefit from key technical advances that have revolutionized general visual tasks.

### 2.2 Hierarchical Paradigms

Inspired by the success of feature pyramid in general computer vision tasks, some methods have attempted to introduce hierarchical structures to WSI analysis, such as HIPT (Chen et al., 2022), H2MIL (Hou et al., 2022), and ZoomMIL (Thandiackal et al., 2022). However, these approaches do not build a feature pyramid organically from a single-scale input as in general computer vision. Instead, they depend on multi-scale inputs, requiring the system to process separate patches from multiple magnification levels (e.g., 5x, 10x, 20x). This strategy introduces significant computational and data management overhead. More importantly, within each scale, these methods still operate isotropically, failing to form a cohesive, end-to-end hierarchical representation. This architectural compromise means the central challenge of building a true feature pyramid from a single-scale input remains largely unaddressed. As a result, WSI analysis has yet to fully harness the powerful and efficient hierarchical architectures that are now state-of-the-art in the broader vision community.

## 3 Method

The core of our backbone is a rulebook-based mechanism: a pre-computed set of instructions that explicitly defines input-output mappings for sparse data. This allows for highly efficient computation by targeting only active features and eliminating redundant operations on empty regions. The SPAN backbone is constructed from a repeating sequence of SAC and CAR modules that adhere to this principle. As illustrated in Fig. 2, the SAC module performs spatial condensation and coarse-grained feature transformation, while the subsequent CAR module employs transformer blocks with shifted windows for fine-grained contextual refinement. This complementary design allows the SPAN backbone to efficiently capture both multi-scale patterns and their long-range dependencies, which can then be utilized by task-specific variants: SPAN-MIL for classification through global token aggregation, and SPAN-UNet for segmentation through hierarchical decoding.

This hierarchical processing repeats with subsequent SAC-CAR modules operating on increasingly condensed features, enabling SPAN to learn pyramid representations that unify multi-granularity information with global understanding. The gradual reduction in spatial resolution also allows SPAN to efficiently manage memory consumption at deeper layers while preserving multi-scale diagnostic patterns.

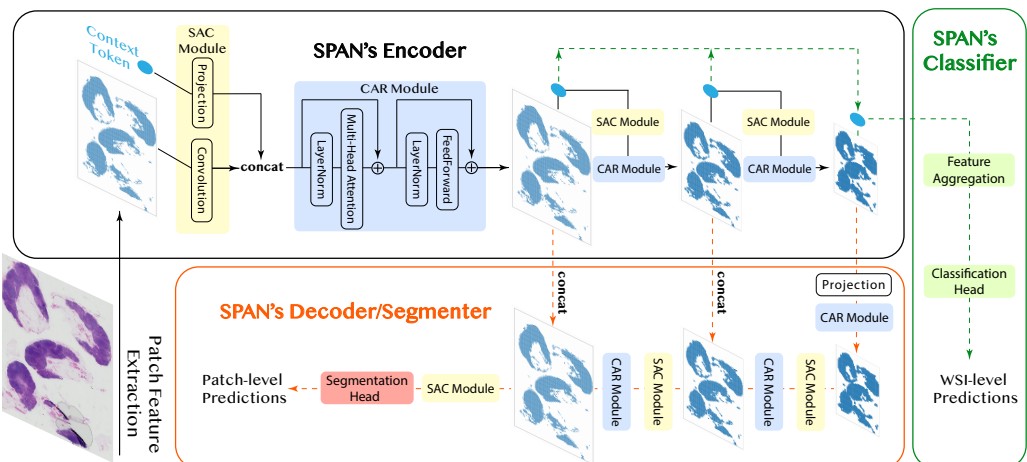

Figure 2: Overall architecture of SPAN. The encoder begins with a SAC module comprising Projection and Convolution components, followed by CAR that employs window attention through LayerNorm, Multi-Head Attention, and Feed-Forward layers for local context modeling. While the initial SAC preserves spatial dimensions with $1 \times 1$ convolution, subsequent SAC modules progressively downsample tokens to approximately 1/4 of their previous token count. This SAC-CAR sequence repeats multiple times for hierarchical feature extraction and refinement. Task-specific paths (dashed lines) enable flexible downstream applications: the decoder/segmenter path utilizes alternating CAR-SAC modules with transposed convolutions in SAC for upsampling and patch-level predictions, while the classifier path employs feature aggregation for WSI-level predictions.

## 3.1 SPATIAL-ADAPTIVE FEATURE CONDENSATION

The SAC module progressively condenses patches into more compact representations through learnable feature transformations. The design of SAC is motivated by two key insights: the inherent multi-scale nature of histopathological diagnosis that pathologists perform, and the computational efficiency required for processing large-scale WSIs. This motivates us to design an adaptive feature extraction process that can handle the irregular spatial distribution of tissue regions.

Our condensation process maintains spatial relationships while progressively reducing spatial dimensions to capture multi-scale patterns. To achieve this efficiently, we implement SAC using sparse convolutions (Liu et al., 2015) for downsampling and hierarchical feature encoding. This choice naturally aligns with the WSI structure, where significant background portions contain uninformative regions, enabling selective computation only where meaningful features are present.

**Sparse Convolution Rulebook** Sparse convolution operations are typically implemented using a rulebook-based approach, which efficiently manages the computation and memory usage for sparse data structures. Specifically, an index matrix $\mathbf{I} = \begin{bmatrix} 1 & 2 & \cdots & N \end{bmatrix}^{\mathrm{T}}$ corresponds to the coordinate matrix $\mathbf{P} = [p_i \mid i \in \mathbf{I}] \in \mathbb{N}^{N \times 2}$ and the feature matrix $\mathbf{X} = [x_i \mid i \in \mathbf{I}] \in \mathbb{R}^{N \times d}$. This structured representation ensures efficient access to coordinates and their associated features during sparse convolution operations.

For each convolutional layer, the output coordinates are computed based on the input coordinates, the kernel size $K$, the dilation $D$, and the layer's stride $S$:

$$\mathbf{P}_{\mathrm{out}} = \left\{ p_{i_{\mathrm{out}}} \mid p_{i_{\mathrm{out}}} = \left\lfloor \frac{p_{i_{\mathrm{in}}} - (K-1) \cdot D}{S} \right\rfloor, \ \forall p_{i_{\mathrm{in}}} \in \mathbf{P}_{\mathrm{in}} \right\}, \quad (1)$$

where $\lfloor \cdot \rfloor$ denotes the floor operation, and $(K-1) \cdot D$ adjusts for the expansion of the receptive field due to the kernel size and dilation. The corresponding output indices $\mathbf{I}_{\mathrm{out}}$ are assigned sequentially starting from 1.

To determine the valid mappings between input and output indices for each kernel offset, we construct a *rulebook* $\mathcal{R}_k$ defined as:

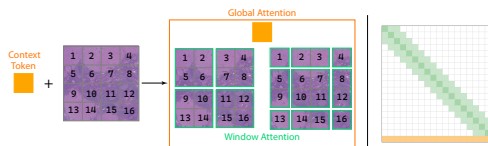

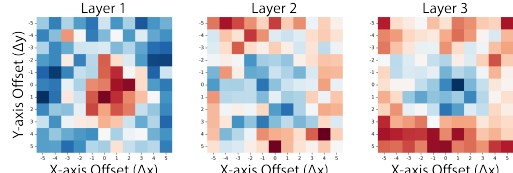

Figure 3: Schematic of CAR. *Left:* The input is partitioned into overlapping $2w \times 2w$ windows. Attention is computed locally within windows (green box) and globally via a learnable token that attends to all tokens (orange box). *Right:* The attention matrix visualizes this: diagonal blocks (green) show local attention, while the full row/column (orange) shows the global token's unrestricted scope.

Figure 4: Layer-wise visualization of learned RPB in SPAN. Each heatmap shows attention bias values as a function of relative positional offsets $(\Delta x, \Delta y)$ between token pairs. Coordinates $(x, y)$ represent the bias when attending to a token at $x$ positions horizontally and $y$ positions vertically relative to the query token. Red and blue indicate higher and lower attention biases, respectively.

$$\mathcal{R}_k = \{(i_{\text{in}}, i_{\text{out}}) \mid p_{i_{\text{in}}} + k = p_{i_{\text{out}}}\}, \quad k \in \mathcal{K}, \tag{2}$$

where $\mathcal{K}$ is the set of kernel offsets, and $p_{i_{\text{in}}}$ and $p_{i_{\text{out}}}$ are input and output coordinates, respectively. Each entry in $\mathcal{R}_k$ represents an atomic operation, specifying that the input position $p_{i_{\text{in}}}$ shifted by the kernel offset $k$ matches the output position $p_{i_{\text{out}}}$. The complete rulebook $\mathcal{R}_{\mathcal{K}} = \bigcup_{k \in \mathcal{K}} \mathcal{R}_k$ efficiently encodes the locations and conditions under which convolution operations are to be performed.

Each sparse convolutional layer performs convolution by executing the atomic operations defined in the rulebook $\mathcal{R}_{\mathcal{K}}$. An atomic operation $(i_{\text{in}}, i_{\text{out}}) \in \mathcal{R}_k$ transforms the input feature $h_{i_{\text{in}}}$ using the corresponding weight matrix $W_l(k)$ and accumulates the result to the output feature $h_{i_{\text{out}}}$. The complete sparse convolution operation for a layer $l$ is defined as:

$$h_{i_{\text{out}}} = \sum_{k \in \mathcal{K}} \sum_{\mathcal{R}_k} W_l(k) h_{i_{\text{in}}} + b_l, \tag{3}$$

where $h_{i_{\text{in}}} \in \mathbb{R}^{d_{\text{in}}}$ is the input feature at index $i_{\text{in}}$, $h_{i_{\text{out}}} \in \mathbb{R}^{d_{\text{out}}}$ is the output feature at index $i_{\text{out}}$, $W_l(k) \in \mathbb{R}^{d_{\text{out}} \times d_{\text{in}}}$ is the weight matrix associated with kernel offset $k$, and $b_l \in \mathbb{R}^{d_{\text{out}}}$ is the bias term for layer $l$.

By utilizing this rulebook-based approach, the sparse convolutional layer efficiently aggregates information from neighboring input features by performing computations only at the necessary locations. This method effectively captures local spatial patterns in the sparse data while significantly reducing computational overhead and memory usage compared to dense convolution operations, as it avoids unnecessary calculations in empty or uninformative regions. For the context token, we compute and average features with all kernel weights and biases if dimension reduction is needed. Otherwise, we maintain an identity projection.

## 3.2 CONTEXT-AWARE FEATURE REFINEMENT

The CAR module builds upon the condensed feature representation to model comprehensive contextual relationships. While the preceding SAC module efficiently captures hierarchical features through progressive condensation, the refined understanding of histological patterns requires modeling both local tissue structures and their long-range dependencies. This dual modeling requirement motivates us to adopt attention mechanisms, which excel at capturing both local and long-range dependencies through learnable interactions between features.

To effectively implement the CAR module, we face several technical challenges in applying attention mechanisms to WSI analysis. Traditional sparse attention approaches (Liu et al., 2021; Beltagy et al., 2020; Zaheer et al., 2020), despite their success in various domains, operate on dense feature matrices by striding over fixed elements in the matrix's memory layout. This approach requires densifying our sparse WSI features and applying padding operations to match the fixed memory layout. Given the high feature dimensionality characteristic of WSI analysis, such transformation would introduce substantial memory and computational overhead while compromising the efficiency established in the previous SAC module. Therefore, we develop a sparse attention rulebook that

directly operates on the sparse feature representation, maintaining compatibility with the SAC module's index-coordinate system. Our approach leverages $\mathbf{I}$ and $\mathbf{P}$ inherited from previous layers to define sparse attention windows, where features within each window can attend to each other without dense transformations. This design preserves both computational efficiency and the sparse structure compatibility established in earlier modules.

**Sparse Attention Rulebook** To efficiently handle sparse data representations, we formulate attention computation using rulebooks following the paradigm of sparse convolutions. The first step is to generate attention windows that define which tokens should attend to each other. For efficient window generation, we temporarily densify $\mathbf{I} \in \mathbb{N}^N$ into a regular grid using patch coordinates $\mathbf{P} \in \mathbb{N}^{N \times 2}$ with zero padding. This enables efficient block-wise memory access on a low-dimensional index matrix rather than operating on a high-dimensional feature matrix. As illustrated in Fig. 3, we stride over the densified index matrix to generate regular and shifted windows, where the shifting operation ensures comprehensive coverage of local contexts. The resulting $\mathcal{W}$ is a collection of windows, where each window contains a set of patch indices excluding padded zeros. These windows effectively define the grouping of indices for constructing an attention rulebook.

To enhance the model's ability to capture global dependencies, we introduce a learnable global context token that provides a shared context accessible to all other tokens. The combined hidden features can be represented as $\mathbf{H} = [h_{i_1}^\top, h_{i_2}^\top, \ldots, h_{i_N}^\top, h_g^\top] \in \mathbb{R}^{(N+1) \times d_{\text{out}}}$, where $h_g$ denotes the global context token. For self-attention computation, we project $\mathbf{H} \in \mathbb{R}^{(N+1) \times d}$ into $\mathbf{Q}$, $\mathbf{K}$, and $\mathbf{V}$ using linear projections.

Having defined the attention windows, we now construct two types of rulebooks to capture both local and global dependencies. For local attention, the rulebook $\mathcal{R}_w$ for each window is defined as:

$$\mathcal{R}_w = \{(i,j) \mid i,j \in w\}, \quad w \in \mathcal{W}, \tag{4}$$

where $\mathcal{W}$ denotes the set of all attention windows, and $i$ and $j$ represent the indices of the input and output patches within the window $w$, respectively. Each entry $(i,j) \in \mathcal{R}_w$ represents a local attention atomic operation between tokens $i$ and $j$. These atomic operations are defined by the following equations. The attention scores are computed with a learnable relative positional bias to account for spatial relationships:

$$e_{ij}^{\text{local}} = \frac{\mathbf{q}_i^\top \mathbf{k}_j}{\sqrt{d}} + B(p_i - p_j), \tag{5}$$

where $\mathbf{q}_i$ and $\mathbf{k}_j$ represent the query and key vectors for local tokens $i$ and $j$, respectively, and $p_i$ and $p_j$ denote their positions. $B(p_i - p_j)$ represents the learnable relative positional biases (RPB) (Liu et al., 2021), parameterized by a matrix $B \in \mathbb{R}^{(2w_{size}-1) \times (2w_{size}-1) \times \text{num\_heads}}$.

The choice of positional encoding is crucial for capturing spatial relationships in WSI analysis. RPB enhances the model's ability to recognize positional nuances and disrupt the permutation invariance inherent in self-attention mechanisms while maintaining parameter efficiency. Alternative approaches present different trade-offs: absolute positional encoding (APE) (Dosovitskiy et al., 2021) would significantly increase the parameter count given the extensive spatial dimension of possible positions in WSIs, while Rotary Position Embedding (RoPE) (Heo et al., 2024; Su et al., 2024) and Attention with Linear Biases (Alibi) (Press et al., 2022), despite their parameter efficiency in language models, prove less effective at capturing spatial relationships in our context.

The final output of the local attention is then computed as:

$$\mathbf{h}_i^{\text{local}} = \sum_{w \in \mathcal{W}} \sum_{j:(i,j) \in \mathcal{R}_{\text{w}}} \frac{\exp(e_{ij}^{\text{local}})}{\sum_{k:(i,k) \in \mathcal{R}_{\text{local}}} \exp(e_{ik}^{\text{local}})} \mathbf{v}_j. \tag{6}$$

To complement local attention with global context modeling, we introduce global attention that operates on all patch tokens and the learnable global context token. The global attention rulebook is defined as:

$$\mathcal{R}_g = \{(i,j),(j,i) \mid i \in [1,N], j \in \{N+1\}\}. \tag{7}$$

The global attention mechanism employs similar formulations as equations equation 5 and equation 6 but excludes the positional bias term, yielding $\mathbf{h}_i^{global}$. While local attention is constrained to windows, global attention spans across the entire feature map through the global context token, enabling comprehensive contextual integration. The final output features combine both local and global dependencies through:

$$\mathbf{h}_i^{\text{out}} = \mathbf{h}_i^{local} + \mathbf{h}_i^{global}. \tag{8}$$

For downstream tasks, SPAN serves s a backbone that support task-specific variants: **SPAN-MIL** employs global token aggregation for slide-level classification tasks, while **SPAN-UNet** utilizes a U-Net-style decoder for patch-level segmentation tasks (implementation details in Appendix B.1).

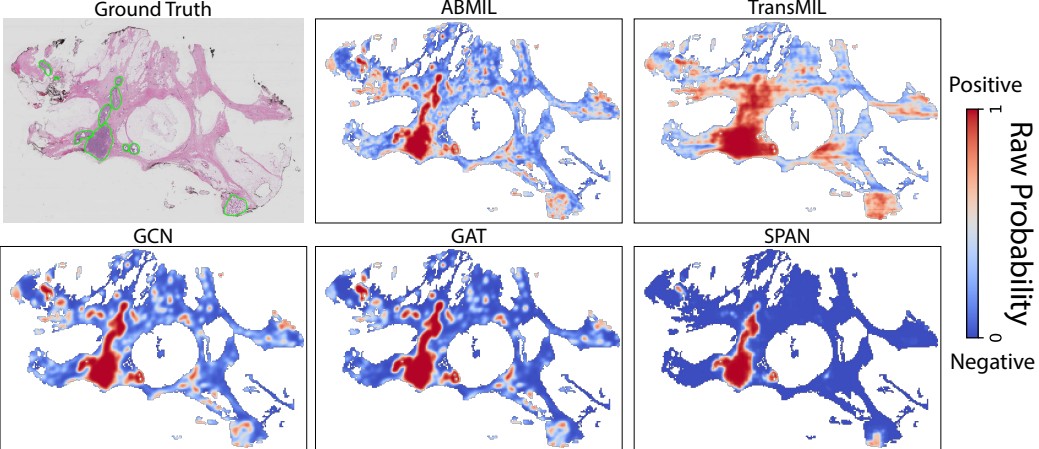

Figure 5: Qualitative comparison of tumor segmentation performance on the unseen test set. The Ground Truth panel depicts the expert-annotated tumor regions enclosed by green contours. The heatmap indicates the predicted probability of tumor presence for each region.

## 4 EXPERIMENTS

We evaluate SPAN across multiple classification and segmentation tasks on public datasets using two feature extractors. ResNet50 ($\sim$ 6 GFLOPS), a long-standing backbone in WSI analysis that continues to be used for its efficiency in immediate deployment and fast prototyping. Virchow2 (Zimmermann et al., 2024) ($\sim$ 360 GFLOPS), a recent domain-specific foundation model that trades 60× more computation for higher accuracy. Detailed experimental setup and implementation details are provided in the Appendix.

Tables 1 and 3 show that both SPAN-MIL and SPAN-UNet consistently achieve SOTA performance across all tasks, demonstrating superior slide-level and patch-level representation learning capabilities. Notably, this strong performance is achieved with a simple cross-entropy loss, whereas competing methods rely on additional auxiliary losses and sophisticated training strategies. This simplicity suggests substantial headroom for further improvements in the SPAN-based models, while competing approaches may have reached a complexity ceiling with diminishing returns for additional modifications. This success stems from undistorted hierarchical spatial encoding that preserves precise patch relationships, coupled with intrinsic multi-level aggregation for classification and a U-Net-like decoding architecture for segmentation. This architecture allows the model to effectively leverage multi-scale contextual information for precise spatial localization, as illustrated in the qualitative examples in Fig. 5.

SPAN's reliability is further highlighted by its consistent performance gains with pathology-specific Virchow2 features, in contrast to baselines that show inconsistent or degraded results. This suggests that SPAN's design becomes more effective when leveraging rich, domain-specific semantic information.

To understand the model's internal mechanics, we visualized the learned relative position bias (RPB) in Fig. 4. The patterns reveal a clear evolution from local attention in early layers to broad, long-

Table 1: Classification performance across CAMELYON16, Yale-HER2, and BRACS datasets

| | CAMELYON16 Dataset | | | | | |
| --- | --- | --- | --- | --- | --- | --- |
| Method | General ResNet50 Feature | | | Pathology-specific Virchow2 Feature | | |
| | Accuracy | AUC | F1 Score | Accuracy | AUC | F1 Score |
| **ABMIL backbone** | | | | | | |
| ABMIL | 0.857 ± 0.085 | 0.915 ± 0.059 | 0.850 ± 0.088 | 0.990 ± 0.015 | 0.999 ± 0.001 | 0.989 ± 0.017 |
| CLAM-SB | 0.873 ± 0.040 | 0.922 ± 0.058 | 0.868 ± 0.039 | 0.983 ± 0.012 | 0.999 ± 0.001 | 0.983 ± 0.012 |
| CLAM-MB | 0.867 ± 0.031 | 0.932 ± 0.023 | 0.862 ± 0.031 | 0.987 ± 0.014 | 0.999 ± 0.001 | 0.986 ± 0.014 |
| DTFD | 0.877 ± 0.073 | 0.947 ± 0.039 | 0.868 ± 0.057 | 0.983 ± 0.020 | 0.994 ± 0.009 | 0.982 ± 0.021 |
| DSMIL | 0.887 ± 0.051 | 0.941 ± 0.025 | 0.881 ± 0.050 | 0.983 ± 0.000 | 1.000 ± 0.001 | 0.983 ± 0.001 |
| MHIM | 0.883 ± 0.053 | 0.929 ± 0.036 | 0.877 ± 0.056 | 0.977 ± 0.025 | 0.999 ± 0.002 | 0.975 ± 0.027 |
| ACMIL | 0.893 ± 0.015 | 0.936 ± 0.023 | 0.889 ± 0.011 | 0.983 ± 0.012 | 1.000 ± 0.001 | 0.983 ± 0.012 |
| **GNN backbone** | | | | | | |
| PatchGCN | 0.833 ± 0.065 | 0.874 ± 0.076 | 0.819 ± 0.072 | 0.979 ± 0.016 | 0.992 ± 0.015 | 0.978 ± 0.017 |
| **Transformer/Mamba backbone** | | | | | | |
| TransMIL | 0.873 ± 0.053 | 0.916 ± 0.056 | 0.867 ± 0.053 | 0.983 ± 0.012 | 1.000 ± 0.001 | 0.983 ± 0.013 |
| RRT | 0.867 ± 0.029 | 0.936 ± 0.038 | 0.862 ± 0.027 | 0.993 ± 0.009 | 1.000 ± 0.001 | 0.993 ± 0.009 |
| MambaMIL | 0.857 ± 0.048 | 0.940 ± 0.038 | 0.848 ± 0.047 | 0.993 ± 0.009 | 1.000 ± 0.001 | 0.993 ± 0.010 |
| **SPAN backbone** | | | | | | |
| SPAN-MIL | 0.903 ± 0.030 | 0.939 ± 0.026 | 0.898 ± 0.032 | 0.993 ± 0.009 | 1.000 ± 0.001 | 0.993 ± 0.010 |
| | Yale-HER2 Dataset | | | | | |
| ABMIL | 0.687 ± 0.084 | 0.778 ± 0.078 | 0.664 ± 0.091 | 0.813 ± 0.038 | 0.857 ± 0.049 | 0.806 ± 0.062 |
| CLAM-SB | 0.713 ± 0.084 | 0.790 ± 0.052 | 0.699 ± 0.090 | 0.793 ± 0.060 | 0.865 ± 0.071 | 0.778 ± 0.064 |
| CLAM-MB | 0.693 ± 0.089 | 0.766 ± 0.105 | 0.684 ± 0.094 | 0.793 ± 0.068 | 0.876 ± 0.055 | 0.784 ± 0.073 |
| DTFD | 0.693 ± 0.086 | 0.764 ± 0.103 | 0.680 ± 0.092 | 0.800 ± 0.085 | 0.860 ± 0.038 | 0.791 ± 0.085 |
| DSMIL | 0.693 ± 0.060 | 0.764 ± 0.041 | 0.676 ± 0.049 | 0.807 ± 0.049 | 0.858 ± 0.042 | 0.793 ± 0.056 |
| MHIM | 0.706 ± 0.104 | 0.744 ± 0.095 | 0.695 ± 0.100 | 0.800 ± 0.108 | 0.872 ± 0.062 | 0.792 ± 0.104 |
| ACMIL | 0.713 ± 0.030 | 0.781 ± 0.059 | 0.685 ± 0.045 | 0.807 ± 0.028 | 0.853 ± 0.032 | 0.787 ± 0.044 |
| PatchGCN | 0.700 ± 0.115 | 0.731 ± 0.100 | 0.690 ± 0.111 | 0.754 ± 0.045 | 0.831 ± 0.033 | 0.689 ± 0.041 |
| TransMIL | 0.672 ± 0.085 | 0.680 ± 0.167 | 0.652 ± 0.113 | 0.807 ± 0.072 | 0.883 ± 0.071 | 0.797 ± 0.081 |
| RRT | 0.647 ± 0.069 | 0.703 ± 0.076 | 0.631 ± 0.072 | 0.753 ± 0.051 | 0.838 ± 0.044 | 0.743 ± 0.049 |
| MambaMIL | 0.717 ± 0.057 | 0.787 ± 0.094 | 0.705 ± 0.059 | 0.717 ± 0.089 | 0.868 ± 0.066 | 0.706 ± 0.095 |
| SPAN-MIL | 0.727 ± 0.072 | 0.786 ± 0.075 | 0.720 ± 0.070 | 0.827 ± 0.086 | 0.888 ± 0.072 | 0.816 ± 0.088 |
| | BRACS Dataset | | | | | |
| Method | General ResNet50 Feature | | | Pathology-specific Virchow2 Feature | | |
| | Accuracy | Macro AUC | Macro F1 | Accuracy | Macro AUC | Macro F1 |
| ABMIL | 0.687 ± 0.023 | 0.828 ± 0.099 | 0.552 ± 0.039 | 0.766 ± 0.020 | 0.897 ± 0.017 | 0.689 ± 0.032 |
| CLAM-SB | 0.687 ± 0.044 | 0.840 ± 0.099 | 0.562 ± 0.041 | 0.757 ± 0.023 | 0.892 ± 0.014 | 0.663 ± 0.028 |
| CLAM-MB | 0.696 ± 0.039 | 0.847 ± 0.085 | 0.545 ± 0.049 | 0.773 ± 0.033 | 0.897 ± 0.015 | 0.698 ± 0.061 |
| DTFD | 0.689 ± 0.027 | 0.828 ± 0.116 | 0.578 ± 0.034 | 0.768 ± 0.015 | 0.884 ± 0.018 | 0.680 ± 0.055 |
| DSMIL | 0.699 ± 0.035 | 0.826 ± 0.101 | 0.553 ± 0.056 | 0.747 ± 0.031 | 0.890 ± 0.018 | 0.643 ± 0.076 |
| MHIM | 0.716 ± 0.028 | 0.847 ± 0.103 | 0.560 ± 0.066 | 0.742 ± 0.020 | 0.887 ± 0.023 | 0.648 ± 0.030 |
| ACMIL | 0.720 ± 0.022 | 0.859 ± 0.085 | 0.604 ± 0.074 | 0.766 ± 0.020 | 0.897 ± 0.017 | 0.689 ± 0.032 |
| PatchGCN | 0.713 ± 0.025 | 0.848 ± 0.101 | 0.610 ± 0.031 | 0.747 ± 0.034 | 0.871 ± 0.028 | 0.662 ± 0.042 |
| TransMIL | 0.692 ± 0.037 | 0.799 ± 0.117 | 0.577 ± 0.034 | 0.754 ± 0.014 | 0.886 ± 0.020 | 0.654 ± 0.052 |
| RRT | 0.718 ± 0.036 | 0.848 ± 0.093 | 0.595 ± 0.065 | 0.761 ± 0.036 | 0.895 ± 0.031 | 0.683 ± 0.062 |
| MambaMIL | 0.706 ± 0.047 | 0.843 ± 0.035 | 0.620 ± 0.059 | 0.771 ± 0.043 | 0.889 ± 0.029 | 0.703 ± 0.049 |
| SPAN-MIL | 0.725 ± 0.038 | 0.853 ± 0.077 | 0.641 ± 0.076 | 0.778 ± 0.028 | 0.898 ± 0.068 | 0.722 ± 0.037 |

range attention in deeper layers. This allows SPAN to dynamically process both fine-grained cellular details and larger tissue architectures, a flexibility not possible with fixed positional encodings.

We conducted ablation studies on the CAMELYON16 dataset with ResNet50 features to validate the contributions of SPAN's components (Table 2, Fig. 6). Aligning with findings in general vision, disabling the SAC module's hierarchical downsampling (via 1x1 convolutions), the CAR module's contextual attention (by setting window size to 0), or the shifted-window mechanism all led to significant performance degradation. Surprisingly, the model performs well even without any positional encoding, possibly due to the rich spatial information inherently captured by its convolution and shift-window attention mechanisms. The inferior performance of Axial RoPE and Alibi likely stems from their fixed distance-decay patterns, which are directly borrowed from other tasks and not optimized for WSI-specific spatial structures. These fixed priors may conflict with the dynamic,

long-range attention that SPAN learns in deeper layers (Fig. 4). For slide-level aggregation, we found that directly using the global context token is simple and effective enough. Finally, as in Fig. 6), increasing the window size beyond a certain point does not necessarily improve performance in our settings; however, it significantly increases memory usage, which may be attributed to insufficient training data to learn complex feature interactions effectively at larger window sizes.

Table 2: Ablations for different settings.

**SPAN-MIL (Slide-level Representation)**

| Configuration | Accuracy | AUC |
|---|---|---|
| *Attention Pooling* | | |
| w/o Context Token | $0.893 \pm 0.037$ | $0.931 \pm 0.031$ |
| w/ Context Token | $0.900 \pm 0.026$ | $0.941 \pm 0.041$ |
| *Positional Encoding* | | |
| Axial Alibi | $0.883 \pm 0.039$ | $0.920 \pm 0.029$ |
| Axial RoPE | $0.880 \pm 0.048$ | $0.917 \pm 0.017$ |
| None | $0.890 \pm 0.019$ | $0.938 \pm 0.027$ |
| *Core Modules* | | |
| No SAC ($K = S = 1$) | $0.879 \pm 0.037$ | $0.928 \pm 0.026$ |
| No CAR ($w_{size} = 0$) | $0.870 \pm 0.022$ | $0.919 \pm 0.038$ |
| No Shifted Window | $0.883 \pm 0.039$ | $0.923 \pm 0.049$ |

**SPAN-UNet (Patch-level Representation)**

| Configuration | Dice | IoU |
|---|---|---|
| *Core Modules* | | |
| No SAC ($K = S = 1$) | $0.826 \pm 0.059$ | $0.708 \pm 0.091$ |
| No CAR ($w_{size} = 0$) | $0.831 \pm 0.056$ | $0.713 \pm 0.083$ |
| *Skip Connection Strategy* | | |
| No Skip Connection | $0.837 \pm 0.059$ | $0.723 \pm 0.088$ |
| w/ Skip Connection (Add) | $0.848 \pm 0.056$ | $0.739 \pm 0.085$ |

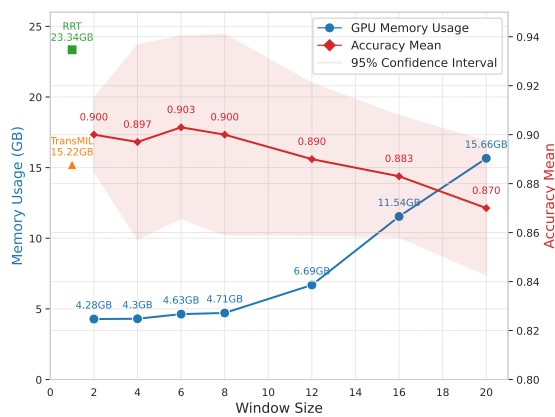

Figure 6: Accuracy and memory usage of SPAN with window sizes from $2 \times 2$ to $20 \times 20$. Each configuration is evaluated over 5 runs, with the mean accuracy and peak memory usage reported.

Table 3: Segmentation performance on histopathology datasets

| Method | CAMELYON16 | | CAMELYON17 | | SegCAMELYON | | BACH | |
|---|---|---|---|---|---|---|---|---|
| | Dice | IoU | Dice | IoU | Dice | IoU | Dice | IoU |
| **General ResNet50 Feature** | | | | | | | | |
| ABMIL[†] | 0.742±0.012 | 0.591±0.016 | 0.548±0.136 | 0.387±0.120 | 0.738±0.038 | 0.586±0.047 | 0.690±0.158 | 0.544±0.181 |
| TransMIL[†] | 0.822±0.051 | 0.700±0.071 | 0.754±0.133 | 0.618±0.156 | 0.818±0.055 | 0.695±0.079 | 0.723±0.176 | 0.588±0.201 |
| RRT[†] | 0.836±0.062 | 0.722±0.094 | 0.786±0.118 | 0.660±0.154 | 0.829±0.066 | 0.712±0.100 | 0.705±0.128 | 0.557±0.159 |
| GCN | 0.841±0.006 | 0.726±0.010 | 0.754±0.080 | 0.610±0.103 | 0.809±0.068 | 0.684±0.098 | 0.695±0.169 | 0.552±0.191 |
| GAT | 0.795±0.029 | 0.661±0.040 | 0.838±0.058 | 0.724±0.087 | 0.805±0.045 | 0.676±0.064 | 0.715±0.136 | 0.571±0.168 |
| SPAN-UNet | 0.885±0.043 | 0.796±0.069 | 0.870±0.038 | 0.771±0.061 | 0.860±0.052 | 0.757±0.080 | 0.783±0.137 | 0.659±0.173 |
| **Pathology-specific Virchow2 Feature** | | | | | | | | |
| ABMIL | 0.809±0.021 | 0.679±0.029 | 0.717±0.087 | 0.565±0.105 | 0.792±0.052 | 0.659±0.069 | 0.702±0.147 | 0.557±0.178 |
| TransMIL | 0.874±0.011 | 0.776±0.017 | 0.878±0.054 | 0.786±0.082 | 0.864±0.035 | 0.762±0.054 | 0.778±0.112 | 0.648±0.145 |
| RRT | 0.876±0.012 | 0.779±0.018 | 0.890±0.032 | 0.803±0.052 | 0.876±0.054 | 0.783±0.084 | 0.748±0.122 | 0.609±0.154 |
| GCN | 0.755±0.070 | 0.611±0.091 | 0.876±0.024 | 0.779±0.038 | 0.809±0.068 | 0.684±0.098 | 0.753±0.121 | 0.615±0.155 |
| GAT | 0.860±0.015 | 0.754±0.024 | 0.853±0.038 | 0.746±0.058 | 0.852±0.066 | 0.747±0.100 | 0.734±0.158 | 0.598±0.194 |
| SPAN-UNet | 0.900±0.013 | 0.818±0.021 | 0.919±0.032 | 0.852±0.053 | 0.884±0.052 | 0.795±0.084 | 0.814±0.096 | 0.695±0.132 |

[†] indicates its corresponding architecture: ABMIL for MLP, TransMIL for vanilla Nystromformer, and RRT for region-based Nystromformer.

Our segmentation ablations further reinforce the adaptation of general vision principles. The results (Table 2) show that our hierarchical pyramid architecture provides a significant performance boost for segmentation tasks, as disabling the core SAC or CAR modules individually resulted in a marked drop in performance. Furthermore, the ablation of skip connections affirms the efficacy of our U-Net-like segmentation design. Removing skip connections for fusing multi-scale features resulted in a clear drop in Dice and IoU scores. Collectively, the consistent validation of these diverse, task-specific principles demonstrates the success and flexibility of our framework in bridging the long-standing gap between general deep learning and computational pathology.

## 5 CONCLUSION

We present SPAN, a sparse-native framework for WSI analysis, bridging general vision principles and computational pathology. SPAN advances WSI modeling by (i) learning hierarchical pyramid representations directly from single-scale inputs, (ii) preserving spatial relationships via spatial-adaptive condensation and context-aware refinement, and (iii) supporting flexible variants for classification and segmentation. Extensive experiments confirm that SPAN delivers consistent gains, establishing it as a WSI backbone that faithfully leverages hierarchical and sparsity-aware biases.

ETHICS STATEMENT

This research focuses on the development of computational pathology methods (SPAN) for analyzing gigapixel whole slide images (WSIs). Our goal is to improve the accuracy and efficiency of histopathological analysis, which can aid in cancer diagnosis, grading, and subtyping.

We exclusively use publicly available and anonymized datasets, ensuring patient privacy is protected as no new patient data was collected for this study.

Our work is intended for research purposes to advance medical image analysis. While SPAN shows promising results, it is not a certified medical device. Any potential clinical application would require rigorous validation and regulatory approval. We envision this method as a decision-support tool for qualified pathologists, not as a replacement for professional medical judgment.

REPRODUCIBILITY STATEMENT

To facilitate reproducibility, we are committed to releasing our code and pretrained models publicly upon acceptance of this paper. We utilized publicly accessible datasets for all experimental work. Comprehensive details regarding our experimental protocol, including dataset information, hyperparameter settings, and training setups, are documented in **Appendix B.2**. This provision is intended to allow other researchers to verify our findings and build upon our work.

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

CONTENTS OF APPENDIX

## A   RELATED WORK

**Self-attention Mechanisms**   The Vision Transformer (ViT) (Dosovitskiy et al., 2021) successfully adapted self-attention mechanisms from NLP (Devlin et al., 2018; Brown et al., 2020) for image recognition. However, its quadratic computational complexity is prohibitive for the tens of thousands of patches generated from a single gigapixel WSI. Subsequent work introduced more efficient variants to handle long sequences. These include models with sparse attention patterns like Longformer (Beltagy et al., 2020) and BigBird (Zaheer et al., 2020), and models with window-based attention like the Swin Transformer (Liu et al., 2021). By computing attention locally within windows and building a hierarchical representation, Swin Transformer achieves linear complexity and captures multi-scale features, leading to state-of-the-art performance on many vision tasks.

Despite these advancements, a fundamental challenge remains in applying these mechanisms to WSIs. They are designed for dense, continuously distributed data. In contrast, the informative patches in WSIs are sparsely and irregularly distributed across a vast, uninformative background. This mismatch makes it inherently difficult to directly apply window-based or dense-matrix-based sparse attention techniques, necessitating specialized approaches that can natively handle sparse data distributions.

**Pyramid Structures in General Visions**   Multi-scale feature representation is a cornerstone of modern computer vision. In CNNs, this is achieved through progressive downsampling (He et al., 2016) and explicit pyramid architectures that capture context at multiple resolutions, such as SPP-Net (He et al., 2015), FPN (Lin et al., 2017), and HRNet (Wang et al., 2020). This powerful paradigm is successfully integrated into vision transformers as well. Models like Pyramid Vision Transformer (PVT) (Wang et al., 2021) and Swin Transformer (Liu et al., 2021) incorporate hierarchical designs with efficient attention, proving the value of multi-scale feature learning for achieving state-of-the-art results.

However, these successful pyramid structures are all designed for dense and uniformly distributed data. They rely on regular downsampling operations (e.g., strided convolutions or patch merging) that are fundamentally inappropriate for the sparse and irregular spatial layout of WSIs. The unique challenges posed by vast uninformative regions prevent the direct application of general-purpose pyramid architectures, leaving a critical gap in WSI analysis.

## B   IMPLEMENTATION AND EXPERIMENTAL DETAILS

### B.1   TASK-SPECIFIC VARIANTS IMPLEMENTATION DETAILS

#### B.1.1   SPAN-MIL: CLASSIFICATION HEAD

We utilize the global context tokens introduced in the CAR module for their comprehensive representations of the WSI across different scales. Let $\mathbf{h}_l^g \in \mathbb{R}^d$ denote the global context token from layer $l \in \{1, \dots, L\}$. The slide-level representation is computed by:

$$\mathbf{h}^{\mathrm{cls}} = \sum_{l=1}^{L} \mathbf{h}_l^g. \tag{9}$$

The classification prediction is obtained through:

$$\hat{y} = \mathrm{softmax}(W^{\mathrm{cls}}\mathbf{h}^{\mathrm{cls}} + b^{\mathrm{cls}}), \tag{10}$$

where $W^{\mathrm{cls}} \in \mathbb{R}^{c \times d}$ and $b^{\mathrm{cls}} \in \mathbb{R}^c$ are learnable parameters, and $c$ is the number of classes.

#### B.1.2   SPAN-UNET: SEGMENTATION HEAD

SPAN naturally extends to a U-Net (Ronneberger et al., 2015) architecture through its hierarchical sparse design. The decoder maintains architectural symmetry with the encoder, using sparse deconvolution for upsampling in place of the downsampling operations.

Let $\{\mathbf{H}_1, \mathbf{H}_2, \ldots, \mathbf{H}_L\}$ denote the multi-scale feature maps from the encoder, where $\mathbf{H}_l \in \mathbb{R}^{N_l \times d}$ represents features at the $l$-th level.

The decoder generates features $\{\mathbf{G}_1, \mathbf{G}_2, \ldots, \mathbf{G}_L\}$, processed at each stage through:

$$\mathbf{G}_l = \text{SAC}(\text{CAR}(\mathbf{X}_l)) \in \mathbb{R}^{N_l \times d}. \tag{11}$$

For the first decoding stage, $\mathbf{X}_1 = \mathbf{H}_L$. For subsequent stages, we implement skip connections by concatenating upsampled features with corresponding encoder features:

$$\mathbf{X}_l = \mathbf{G}_{l-1} \, \| \, \mathbf{H}_{L-l+1} \in \mathbb{R}^{N_l \times 2d}, \tag{12}$$

where $\|$ denotes feature concatenation. The final segmentation prediction at position $i$ is:

$$\hat{y}_i = \text{softmax}(W^{\text{seg}} \mathbf{G}_L[i] + b^{\text{seg}}), \tag{13}$$

where $W^{\text{seg}} \in \mathbb{R}^{s \times d}$ and $b^{\text{seg}} \in \mathbb{R}^s$ are learnable parameters, and $s$ is the number of segmentation classes.

## B.2 EXPERIMENTAL SETUP

### B.2.1 CLASSIFICATION DATASETS

WSI classification involves automatically categorizing tissues based on histopathological features, an essential process for accurate diagnosis, grading, and personalized treatment planning. We assessed SPAN's classification performance on three distinct diagnostic tasks, specifically tumor detection using the CAMELYON16 dataset (Bejnordi et al., 2017), tumor grading employing the BRACS dataset (Brancati et al., 2022), and HER2 biomarker status prediction using the Yale-HER2 dataset Farahmand et al. (2022).

We followed the same strategy as above: all available slides were pooled, randomly shuffled, and split into training ($\sim$70%), validation ($\sim$15%), and test ($\sim$15%). Experiments were repeated under five random seeds (0–4). Model selection is based on validation set performance. Crucially, final predictions are made via direct class probability argmax, without any post-hoc threshold optimization, to better mirror real-world clinical deployment scenarios.

### B.2.2 SEGMENTATION DATASETS

Slide-level segmentation requires precise pixel-level delineation of tumor regions, a challenging task crucial for diagnosis and prognosis. To rigorously evaluate SPAN's performance, we used fully annotated slides from multiple datasets: SegCAMELYON, Yale-HER2 (Farahmand et al., 2022), and BACH (Aresta et al., 2019). To construct the SegCAMELYON benchmark, we curated tumor-positive slides from CAMELYON16 (Bejnordi et al., 2017) and CAMELYON17 (Bandi et al., 2018), applied exclusion masks to remove ambiguous regions, and consolidated the processed samples into a unified dataset.

All available slides were pooled, randomly shuffled, and split into training ($\sim$70%), validation ($\sim$10%), and test ($\sim$20%). Experiments were repeated under five random seeds (0–4) to ensure robustness. Patches with over 20% tumor area are labeled positive for patch-level ground truth generation. For segmentation, we adopted 3-layer GCN and GAT models with 8-adjacent connectivity, following standard WSI analysis practices (Hou et al., 2022; Chen et al., 2021; Wu et al., 2023). Model selection is based on validation set performance. Crucially, final predictions are made via direct class probability argmax, without any post-hoc threshold optimization, to better mirror real-world clinical deployment scenarios.

For segmentation training, we employed a hybrid loss that combines cross-entropy (CE) and Dice loss. Specifically, given the predicted probability map $\mathbf{p}$ and the ground-truth mask $\mathbf{y}$, we compute the standard pixel-wise CE loss $\mathcal{L}_{\text{CE}}(\mathbf{p}, \mathbf{y})$ and the Dice loss $\mathcal{L}_{\text{Dice}}(\mathbf{p}, \mathbf{y})$. The final objective is defined as

$$\mathcal{L} = \begin{cases} (1 - \lambda) \, \mathcal{L}_{\text{CE}} + \lambda \, \mathcal{L}_{\text{Dice}}, & \text{if } \sum \mathbf{y} > 0, \\ \mathcal{L}_{\text{CE}}, & \text{otherwise,} \end{cases}$$

where $\lambda = 0.75$ is the Dice weight. This design follow common practices in computer vision community, encouraging accurate boundary delineation when positives are present. All baseline methods were trained under this unified loss function for fair comparison.

### B.2.3 SLIDE PREPROCESSING

Our preprocessing pipeline extends CLAM (Lu et al., 2021) by adding a grid alignment step, adjusting patch boundaries to the nearest multiple of 224 pixels for precise spatial coordinates.

To evaluate feature-space adaptability, we used two pre-trained encoders to generate patch-level features from all datasets at 20x magnification. All patches were resized to 224×224 pixels prior to feature extraction. Our preprocessing pipeline addresses coordinate inconsistencies that arise from CLAM's background filtering mechanism. The original CLAM pipeline can generate patches with irregular starting coordinates due to tissue contour boundaries, making it difficult to establish consistent spatial relationships in a regular grid system. To resolve this, we introduced a grid alignment step that extends tissue contours to align with 224×224 pixel boundaries before patch extraction.

---

**Algorithm 1:** Expand Contours

**global** step_size = 224
**def** *extend_contour(start_x, start_y, w, h)*:
   w += start_x % step_size
   h += start_y % step_size
   start_x -= start_x % step_size
   start_y -= start_y % step_size
   **return** start_x, start_y, w, h
# contour: (start_x, start_y, w, h)
contour = extend_contour(contour)

---

This alignment ensures that all patches map precisely to a regular grid coordinate system, eliminating potential rounding errors in spatial relationship modeling.

### B.2.4 PATCH FEATURE EXTRACTOR

In all experiments, the weights of these encoders were kept frozen to ensure a consistent feature extraction process.

**ResNet50** As a standard baseline, we used a ResNet50 model pre-trained on ImageNet (He et al., 2016). Following common practice in WSI analysis, we removed the final fully connected classification layer and used the output of the global average pooling layer. This process yields a 1024-dimensional feature vector for each patch, representing general-purpose visual features learned from natural images.

**Virchow2** (Zimmermann et al., 2024), a massive pan-cancer collection of over 1.5 million WSIs and associated medical texts. This self-supervised training on domain-specific data allows Virchow2 to learn representations that are highly attuned to histopathological nuances.

---

**Algorithm 2:** SPAN Backbone with Rulebook Mechanism

---

**Input:** $\mathbf{P} \in \mathbb{N}^{N \times 2}$ (coordinates), $\mathbf{X} \in \mathbb{R}^{N \times d}$ (features)
**Output:** Refined features and global context
**for** *each layer in backbone* **do**

    // SAC Module:  Sparse Convolution Rulebook
    $\mathbf{P}_{\text{out}} \leftarrow$ compute_output_coords($\mathbf{P}$, $K$, $S$, $D$)
    $\mathcal{R}_{\text{sparse}} \leftarrow$ build_sparse_rulebook($\mathbf{P}$, $\mathbf{P}_{\text{out}}$, $\mathcal{K}$)
    $\mathbf{X} \leftarrow$ execute_sparse_conv($\mathbf{X}$, $\mathcal{R}_{\text{sparse}}$, $\mathbf{W}$)
    // CAR Module:  Sparse Attention Rulebook
    $\mathcal{W} \leftarrow$ generate_windows($\mathbf{P}_{\text{out}}$, window_size)
    $\mathcal{R}_{\text{local}} \leftarrow \{(i,j) \mid i,j \in w, \forall w \in \mathcal{W}\}$
    $\mathcal{R}_{\text{global}} \leftarrow \{(i, N+1), (N+1, i) \mid i \in [1, N]\}$
    $\mathbf{X} \leftarrow$ execute_attention($\mathbf{X}$, $\mathcal{R}_{\text{local}}$, $\mathcal{R}_{\text{global}}$)
    $\mathbf{P} \leftarrow \mathbf{P}_{\text{out}}$

**return** $\mathbf{X}$*, global_token*

---

**Algorithm 3:** Build Sparse Attention Rulebook

---

**Input:** $\mathbf{P} \in \mathbb{N}^{N \times 2}$ (coordinates), $w$ (window size)
**Output:** $\mathcal{R}_{\text{local}}, \mathcal{R}_{\text{global}}$ (attention rulebooks)
// Create coordinate hash mapping
hash_ids $\leftarrow$ arange(1, $N+1$)
coord_transpose $\leftarrow \mathbf{P}$.transpose()
spatial_bounds $\leftarrow$ (max(coord_transpose[0]) + 1, max(coord_transpose[1]) + 1)
coord_tensor $\leftarrow$ create_sparse_coo(coord_transpose, hash_ids, spatial_bounds)
index_matrix $\leftarrow$ coord_tensor.to_dense()
// Generate attention windows via spatial indexing
**if** *index_matrix.size() $< 2w \times 2w$* **then**

    // Compact space:  full attention
    spatial_indices $\leftarrow$ arange(num_elements)
    query_idx $\leftarrow$ spatial_indices.repeat_interleave(num_elements)
    key_idx $\leftarrow$ spatial_indices.repeat(num_elements)

**else**

    // Extended space:  windowed attention
    window_blocks $\leftarrow$ generate_windows(index_matrix, $w$, mode)
    block_capacity $\leftarrow (2w)^2$
    intra_indices $\leftarrow$ arange(block_capacity)
    query_idx $\leftarrow$ intra_indices.unsqueeze(1).repeat(1, block_capacity).flatten()
    key_idx $\leftarrow$ intra_indices.repeat(block_capacity)
    query_hash $\leftarrow$ window_blocks.flatten()[query_idx]
    key_hash $\leftarrow$ window_blocks.flatten()[key_idx]

// Filter valid mappings and normalize hash indices
valid_mask $\leftarrow$ (query_hash $\neq 0$) $\wedge$ (key_hash $\neq 0$) $\wedge$ (query_hash $\neq$ key_hash)
$\mathcal{R}_{\text{local}} \leftarrow$ (query_hash[valid_mask] - 1, key_hash[valid_mask] - 1)
// Global context rulebook
$\mathcal{R}_{\text{global}} \leftarrow \{(\alpha, N+\beta), (N+\beta, \alpha) \mid \alpha \in [0, N-1], \beta \in [0, \text{num\_ctx} - 1]\}$
**return** $\mathcal{R}_{local}, \mathcal{R}_{global}$

---

---

**Algorithm 4:** Spatial Window Indexing

---

**Input:** index_matrix, $w$ (window radius), mode
**Output:** Active window blocks
$h, width \leftarrow$ index_matrix.size()
// Compute spatial alignment padding
row_align $\leftarrow (2w - h \bmod 2w) \bmod 2w$
col_align $\leftarrow (2w - width \bmod 2w) \bmod 2w$
**if** *row_align > 0 or col_align > 0* **then**
    ⌊ index_matrix $\leftarrow$ spatial_pad(index_matrix, alignment_spec, mode)
// Efficient spatial tessellation
window_tessellation $\leftarrow$ index_matrix.unfold(0, $2w$, $2w$).unfold(1, $2w$, $2w$)
// Filter active windows by occupancy
occupancy_map $\leftarrow$ window_tessellation.sum(dim=[-2, -1])
**return** *window_tessellation[occupancy_map > 0]*

---

---

**Algorithm 5:** Execute Rulebook-based Attention

---

**Input:** $\mathbf{Q}, \mathbf{K}, \mathbf{V}$ (projections), $\mathcal{R}_{\text{local}}, \mathcal{R}_{\text{global}}$ (rulebooks)
**Output:** $\mathbf{H}_{\text{out}}$ (refined features)
// Local attention via spatial rulebook
**for** $(\alpha, \beta) \in \mathcal{R}_{local}$ **do**
    ⌊ $\phi_{\alpha\beta} \leftarrow \frac{\mathbf{q}_\alpha^\top \mathbf{k}_\beta}{\sqrt{d}} + \mathcal{B}(\mathbf{P}[\alpha] - \mathbf{P}[\beta])$
$\mathbf{H}_{\text{local}} \leftarrow$ apply_rulebook_softmax($\{\phi_{\alpha\beta}\}, \mathbf{V}, \mathcal{R}_{\text{local}}$)
// Global attention via context rulebook
**for** $(\alpha, \beta) \in \mathcal{R}_{global}$ **do**
    ⌊ $\psi_{\alpha\beta} \leftarrow \frac{\mathbf{q}_\alpha^\top \mathbf{k}_\beta}{\sqrt{d}}$
$\mathbf{H}_{\text{global}} \leftarrow$ apply_rulebook_softmax($\{\psi_{\alpha\beta}\}, \mathbf{V}, \mathcal{R}_{\text{global}}$)
$\mathbf{H}_{\text{out}} \leftarrow \mathbf{H}_{\text{local}} + \mathbf{H}_{\text{global}}$
**return** $\mathbf{H}_{out}$

---

