# OpenReview forum: "Learning Pyramid Representations from Gigapixel Histopathological Images"
_ICLR.cc/2026/Conference — ICLR 2026 Conference Withdrawn Submission_

### Official Review · Reviewer_cmyf · 2025-10-30

**Soundness:** 2
**Presentation:** 2
**Contribution:** 2
**Rating:** 2
**Confidence:** 4

**Summary:**

The paper describes a hierarchical framework for analysis of histopathology images. Authors propose a Sparse Pyramid Attention mechanism integrating a spatial adaptive condensation module and a context aware feature refinement module. Authors assess their method on various benchmarks related to slide classification and segmentation against a variety of baselines.

**Strengths:**

1. Paper well written and clear. I enjoyed reading it, and it is generally well organized.
2. Methodology described with enough details and technically sound.
3. Extensive assessment and benchmarking against competing methods.

**Weaknesses:**

1. Limited technical novelty:  despite the conceptual definition of sparse attention rulebook, that appears like an incremental contribution, the usage of pyramid attention
has been already proposed in the literature:
Yu, Yang, et al. "Multi-scale spatial pyramid attention mechanism for image recognition: An effective approach." Engineering Applications of Artificial Intelligence 133 (2024): 108261.
Hua, X., Xiang, G., Yuan, H., Zou, L., Wang, L., & Hong, H. (2025). An Efficient and Lightweight Pyramid Attention for Image Deblurring. Pattern Recognition, 112506.

2. Positioning with respect to SOTA: Authors fail in providing a convincing discussion of related work and detailing the actual contributions, and I find unfortunate that a short related work section is provided in the literature, and the critical discussion of relevant methods is scattered here and there in the introduction and various parts.
I don't find particularly convincing the claim in the introduction
"For example, TransMIL (Shao et al.,
2021) relies on re-squaring with Nystr ̈om attention and [CLS] tokens, while others introduce region
attention after dense reshaping (Tang et al., 2024). These approaches inevitably distort positional
information and restrict modeling to isotropic representations, failing to exploit the hierarchical
structures that have proven vital in general computer vision"  while in the result section the reported improvement of SPAN with respect to these methods appear  slightly incremental.

3. Limited Assessment: only few qualitative results are provided and showcasing limitations of the method (in Fig. 5 the detected neoplastic areas from SPAN appear underestimated, and this would be a strong limitation from diagnostic perspective). Also, it is not clear at what level of zoom the method works: authors mention 20X magnification, but it is not clear how their framework would work at nuclear level (I am not sure that they managed to consider the highest magnification level of slides). Finally, apart of memory consuption, no information is provided about efficiency, especially in comparison with baselines. How much does it take to train? How much for inference?

**Questions:**

1. Provide more details about architectural choice, and training of the model
2. Provide more qualitative results, and how the framework works at highest magnification level (nuclear).
3. Reorganize the related work section in the main manuscript, and position the proposed framework with respect to SOTA.

**Details Of Ethics Concerns:**

No concerns.

---

### Official Review · Reviewer_kHnC · 2025-10-31

**Soundness:** 3
**Presentation:** 3
**Contribution:** 2
**Rating:** 4
**Confidence:** 5

**Summary:**

This paper introduces SPAN, a "sparse-native" framework for analyzing gigapixel WSIs. It aims to address a key challenge where previous methods distort spatial context by either discarding patch shapes or forcibly reshaping them. SPAN, in contrast, preserves the exact spatial relationships of tissue patches and builds a hierarchical pyramid representation from a single-scale input to improve WSI classification and segmentation.

**Strengths:**

1.	The proposed method directly processes irregular tissue patches, preserving their exact spatial relationships rather than distorting them by reshaping.

2.	It builds a hierarchical pyramid representation directly from a single-scale input, unlike other approaches that require multiple magnification levels.

**Weaknesses:**

1.	A major part of Section 3 (Method) describes well-established methods, such as Sparse Conv, local window attention, etc., indicating that most of the proposed mechanism is established techniques from prior research in the field. This makes the approach in this work more of an engineering assembly than a methodological innovation.

2.	Please mark the best and second-best models in Tables 1,2, and 3. The current version shows poor readability.

3.	The CAMELYON16 dataset loses its significance as a benchmark because all models can achieve near-perfect scores (with Virchow2 or other pathology foundation models as the feature extractor). Thus, it only shows the strong capability of Virchow2, instead of the proposed method.

4.	Moreover, the usage of ResNet50 as the feature extractor does not make much sense since there are many pathology foundation models available, and they are much better than ResNet50.

5.	Several highly related papers are not properly cited or compared:

$\quad$ [1] (Hierarchical Pooling / Condensing) Guo, Zhengrui, et al. "Histgen: Histopathology report generation via local-global feature encoding and cross-modal context interaction." International Conference on Medical Image Computing and Computer-Assisted Intervention. Cham: Springer Nature Switzerland, 2024.

$\quad$ [2] (Spatial-aware MIL) Yang, Zekang, Hong Liu, and Xiangdong Wang. "Scmil: Sparse context-aware multiple instance learning for predicting cancer survival probability distribution in whole slide images." International Conference on Medical Image Computing and Computer-Assisted Intervention. Cham: Springer Nature Switzerland, 2024.

$\quad$ [3] (Context-aware long sequence modeling) Guo, Zhengrui, et al. "Context matters: Query-aware dynamic long sequence modeling of gigapixel images." arXiv preprint arXiv:2501.18984 (2025).

$\quad$ [4] (Spatial-aware Aggregation) Xu, Hanwen, et al. "A whole-slide foundation model for digital pathology from real-world data." Nature 630.8015 (2024): 181-188.

$\quad$ [5] (Local Attention / Long sequence modeling) Li, Honglin, et al. "Rethinking transformer for long contextual histopathology whole slide image analysis." Advances in Neural Information Processing Systems 37 (2024): 101498-101528.

$\quad$ [6] (Pathology Image Segmentation) Chen, Zhixuan, et al. "Segment Anything in Pathology Images with Natural Language." arXiv preprint arXiv:2506.20988 (2025).

**Questions:**

1.	For the processed WSI (after feature extraction, before feeding to the first SAC module), how did the authors maintain its spatial structure in the first place? And does it contain background tokens (or is it solely the tissue tokens)?

2.	Could the authors please use other datasets than CAMELYON16 for comparison?

3.	Could the authors please use other feature extractors, such as UNI [1], GPFM [2], instead of ResNet50?

$\quad$ [1] (Pathology Foundation Model) Chen, Richard J., et al. "Towards a general-purpose foundation model for computational pathology." Nature medicine 30.3 (2024): 850-862.

$\quad$ [2] (Pathology Foundation Model) Ma, Jiabo, et al. "A generalizable pathology foundation model using a unified knowledge distillation pretraining framework." Nature Biomedical Engineering (2025): 1-20.

4.	The authors mentioned “This success stems from undistorted hierarchical spatial encoding that preserves precise patch relationships” in the second paragraph of Section 4. How to prove this? In several datasets, ABMIL based on Virchow2 feature already achieved the best AUC.

5.	How are models like ABMIL, TransMIL, and RRT used for segmentation?

6.	Could the author please cite these papers in the Introduction or Related Work sections, since they are relevant:

$\quad$ [1] (Hierarchical Pooling / Condensing) Guo, Zhengrui, et al. "Histgen: Histopathology report generation via local-global feature encoding and cross-modal context interaction." International Conference on Medical Image Computing and Computer-Assisted Intervention. Cham: Springer Nature Switzerland, 2024.

$\quad$ [2] (Spatial-aware MIL) Yang, Zekang, Hong Liu, and Xiangdong Wang. "Scmil: Sparse context-aware multiple instance learning for predicting cancer survival probability distribution in whole slide images." International Conference on Medical Image Computing and Computer-Assisted Intervention. Cham: Springer Nature Switzerland, 2024.

$\quad$ [3] (Context-aware long sequence modeling) Guo, Zhengrui, et al. "Context matters: Query-aware dynamic long sequence modeling of gigapixel images." arXiv preprint arXiv:2501.18984 (2025).

$\quad$ [4] (Spatial-aware Aggregation / Pathology Foundation Model) Xu, Hanwen, et al. "A whole-slide foundation model for digital pathology from real-world data." Nature 630.8015 (2024): 181-188.

$\quad$ [5] (Local Attention / Long sequence modeling) Li, Honglin, et al. "Rethinking transformer for long contextual histopathology whole slide image analysis." Advances in Neural Information Processing Systems 37 (2024): 101498-101528.

$\quad$ [6] (Pathology Foundation Model) Chen, Richard J., et al. "Towards a general-purpose foundation model for computational pathology." Nature medicine 30.3 (2024): 850-862.

$\quad$ [7] (Pathology Foundation Model) Wang, Xiyue, et al. "A pathology foundation model for cancer diagnosis and prognosis prediction." Nature 634.8035 (2024): 970-978.

$\quad$ [8] (Pathology Foundation Model) Ma, Jiabo, et al. "A generalizable pathology foundation model using a unified knowledge distillation pretraining framework." Nature Biomedical Engineering (2025): 1-20.

$\quad$ [9] (Pathology Image Segmentation) Chen, Zhixuan, et al. "Segment Anything in Pathology Images with Natural Language." arXiv preprint arXiv:2506.20988 (2025).

---

### Official Review · Reviewer_XHtw · 2025-11-04

**Soundness:** 2
**Presentation:** 3
**Contribution:** 2
**Rating:** 4
**Confidence:** 4

**Summary:**

Learning Whole Slide Image (WSI) representation is challenging due to large size/resolution and the sparse distribution of information/regions of interest.  The paper proposes a new method, SPAN, to learn WSI representation using a transformer-style architecture. SPAN incorporates two fundamental ideas: learning hierarchical features corresponding to different resolutions and learning the spatial relationships between patches.
To achieve this, SPAN uses two distinct modules: Spatial-Adaptive Feature Condensation (SAC) and Context-Aware Feature Refinement (CAR). SAC learns hierarchical representations from a single input in an end-to-end fashion, while CAR uses attention to learn local and global context about the patches.
To avoid the quadratic complexity of vanilla attention, the authors propose using a pre-defined sparse attention rulebook. Extensive experiments and comparisons with multiple baselines are done to evaluate the efficacy of the method.

**Strengths:**

* The paper is overall well-written and tries to address an important problem in learning WSI representation.

* Compared to prior, which use image at different resolution to learn multi-resolution features, SPAN can learn multi-resolution feature from a single input in an end-to-end manner.

* The idea of using sparse and convolution rulebook to circumvent high complexity of attention is interesting and novel.

* The method is extensively benchmarked against multiple baselines (Table 1 and 3).

* While the improvement for classification tasks in not significant (explained in the weakness section), SPAN outperforms existing baselines for segmentation task (table 3).

* Ablation in table 2. shows that SAC and CAR modules helps in learning better features.

**Weaknesses:**

* While the method sounds intuitive, the improvement over baselines in Table 1 is not convincing. In particular, for the CAMELYON16 and BRACS datasets, the improvement over the baseline is not significant. For e.g., SPAN-MIL achieves 72.5% acc for the BRACS dataset compared to 72% with ACMIL using ResNet features. Similarly with Virchow2 features, SPAN-MIL achives 77.8% compared to 77.3% with CLAM.

* SPAN does not improve AUC scores for the classification tasks in Table 1. In my opinion, AUC is a more important metric than accuracy, especially for unbalanced datasets.

* Details about the total number of parameters for each baseline in Tables 1 and 3 are missing, making it difficult to understand if the improvement is due to a larger number of parameters or the SPAN methodology.

* It is not clear how the rulebooks are constructed. Providing more implementation details would be helpful in making the methodology easier to understand.

* Similarly, more details about how matrices I and P are constructed are required in the methodology section.

**Questions:**

1. Do you construct different rulebook for different inputs? What's the computational overhead?

2.   > we temporarily densify I ∈ NN into a regular grid using patch coordinates P ∈ NN × 2 with zero padding.

       How do you densify I? Do you use patch selection?

3. Do you have any insights why the method works better for segmentation task and not  for classification tasks? I think understanding how SPAN learns features differently will be important and it might give you insights why AUC scores are not improved (in table 1)

---

### Note · Authors · 2025-11-14

I have read and agree with the venue's withdrawal policy on behalf of myself and my co-authors.